# NOTCH and AKT Signalling Interact to Drive Mammary Tumour Heterogeneity

**DOI:** 10.3390/cancers15174324

**Published:** 2023-08-29

**Authors:** Liliana Ordonez, Giusy Tornillo, Howard Kendrick, Trevor Hay, Matthew John Smalley

**Affiliations:** The European Cancer Stem Cell Research Institute, School of Biosciences, Cardiff University, Hadyn Ellis Building, Maindy Road, Cardiff CF24 4HQ, UK; lili.ordonez@uwe.ac.uk (L.O.); tornillog@cardiff.ac.uk (G.T.);

**Keywords:** mammary tumour, histotype, NOTCH, PTEN, PI3K, AKT, mouse models

## Abstract

**Simple Summary:**

Effective personalised cancer therapy depends on an understanding of the fundamental biological differences between tumours. Such differences may include the activation or suppression of molecular pathways involved in the development and regulation of the normal cells that give rise to the cancer of interest. One such candidate pathway in mammary/breast cancer is NOTCH signalling. In a mouse model of mammary cancer, which normally develops four different histological tumour types upon knockout of the *Pten* and *p53* tumour suppressor genes in the mammary gland, the additional knockout of the *Notch1* or *Notch2* genes did not alter the kinetics of tumour onset but did significantly change the relative proportions of different tumour types. This was accompanied by changes in PI3K/AKT signalling. We suggest PI3K/AKT and NOTCH signalling interact to determine mouse mammary tumour histotype.

**Abstract:**

A better understanding of the mechanisms generating tumour heterogeneity will allow better targeting of current therapies, identify potential resistance mechanisms and highlight new approaches for therapy. We have previously shown that in genetically modified mouse models carrying conditional oncogenic alleles, mammary tumour histotype varies depending on the combination of alleles, the cell type to which they are targeted and, in some cases, reproductive history. This suggests that tumour heterogeneity is not a purely stochastic process; rather, differential activation of signalling pathways leads to reproducible differences in tumour histotype. We propose the NOTCH signalling pathway as one such pathway. Here, we have crossed conditional knockout *Notch1* or *Notch2* alleles into an established mouse mammary tumour model. *Notch1/2* deletion had no effect on tumour-specific survival; however, loss of *Notch* alleles resulted in a dose-dependent increase in metaplastic adenosquamous carcinomas (ASQCs). ASQCs and adenomyoepitheliomas (AMEs) also demonstrated a significant increase in AKT signalling independent of *Notch* status. Therefore, the NOTCH pathway is a suppressor of the ASQC phenotype, while increased PI3K/AKT signalling is associated with ASQC and AME tumours. We propose a model in which PI3K/AKT and NOTCH signalling act interact to determine mouse mammary tumour histotype.

## 1. Introduction

Both inter- and intra-tumour heterogeneity remain a significant challenge to effective cancer therapy. A better understanding of the fundamental mechanisms generating tumour heterogeneity will not only allow a more precise assessment of which treatments are suitable for which cancers, but also identify potential resistance mechanisms and new approaches for therapy.

We have previously demonstrated that mouse mammary tumours developing in *Brca1 p53*, *Brca2 p53* or *Pten p53* loss-of-function models, or in a *Her2/Neu* gain-of-function model, can be grouped into one of four distinct histotypes: adenomyoepithelioma (AME), metaplastic adenosquamous carcinoma (ASQC), metaplastic spindle cell carcinoma (MSCC) and adenocarcinoma of no special type (AC(NST)) [1,2,3]. All four of these histotypes can be seen in human breast cancer. The metaplastic spindle cell tumours and adenosquamous tumours are grouped under the ‘metaplastic breast cancer’ subtype in humans but are rare (<1% of invasive human breast cancer) [4]. Adenomyoepitheliomas are also seen in human breast cancer but are usually (although not always) benign and very rare (case reports and small series only) [4]. Adenocarcinoma of no special type, however, is the equivalent of the most common form of invasive breast cancer in humans, Invasive Breast Carcinoma of no special type [1,4].

The proportion of different tumour types in mouse models varies depending on the combination of alleles, the mammary cell layer to which their loss is targeted and, in some cases, reproductive history. In particular, when tumour formation was targeted to mammary luminal stem/progenitor cells (using the *β-lactoglobulin* promoter-driven CRE (*BlgCre*)), *BlgCre Pten* mice developed mainly AME and ASQC tumours, whereas *BlgCre Pten p53* mice also developed MSCC and AC(NST) histotypes. In contrast, the majority of tumours from *BlgCre Brca1 p53* and *BlgCre Brca2 p53* backgrounds were the MSCC and AC(NST) types [1,2]. Furthermore, in a model in which *BlgCre* activated an *Erbb2*/*HER2* orthologue, tumour histotype varied from AC(NST) in virgin animals to ASQC in parous animals [3]. In the latter model, we demonstrated ASQC formation is associated with the activation of the epidermal differentiation cluster of genes (EDC). This suggests that the spectrum of tumours observed in each model is not a purely stochastic process; rather, we hypothesise that the differential activation of signalling pathways, particularly pathways involved in mammary development [5], leads to reproducible differences in tumour histotype between models.

One such candidate pathway is NOTCH signalling. The mammalian Notch signalling pathway consists of a family of four transmembrane receptors (NOTCH1–4) that, following activation by one of a number of possible ligands tethered to neighbouring cells, undergo multiple cleavage events to release the Notch intracellular domain (NICD). NICD translocates to the nucleus, where it forms part of a transcriptional activating complex [6]. NOTCH signalling is a key pathway in the differentiation of mammalian tissues [7]. In the mammary epithelium, NOTCH1 activation promotes a luminal cell fate [8], while NOTCH3 is expressed in ductal luminal progenitors and is important for the formation of the luminal cell layer [9]. NOTCH4 is important for maintaining ‘stemness’ in breast cancer stem cells [10,11,12]. In breast cancer, elevated *NOTCH1* expression is significantly associated with poor-prognosis breast cancer, while *NOTCH2* expression is associated with good-prognosis breast cancer [11,12,13,14]. However, whether these pathways directly determine whether tumours are more or less aggressive, or are simply associated with tumour histotypes with different survival outcomes but do not directly determine such outcomes, is unclear.

Here, we have crossed conditional knockout *Notch1* or *Notch2* alleles into our *BlgCre Pten p53* GEMM line. *Notch1/2* deletion had no effect on tumour-specific survival; however, loss of *Notch* alleles resulted in a dose-dependent increase in ASQC and AME tumour histotypes. AME and ASQC tumours also demonstrated an increase in AKT signalling independent of *Notch* status. We propose a model in which NOTCH signalling does not directly affect survival, but rather, PI3K/AKT and NOTCH combine to regulate cellular differentiation pathways in mammary tumours and, thus, determine tumour histotype.

## 2. Materials and Methods

### 2.1. Generation of Mouse Cohorts

All animal procedures were carried out according to current UK Home Office regulations following local ethical committee approval by the Institute of Cancer Research and Cardiff University Animal Welfare Ethical Review Bodies and under the authority of the appropriate personal and project licenses. ARRIVE guidelines were followed. Mice were maintained on an outbred, Black 6 (C3H) background and were fed standard diet and water ad libitum.

All cohorts carried the *Cre* transgene under the control of the *Blg* promoter, driving tumour development from mammary luminal stem/progenitor cells [2]. The *Pten^flox/flox^ p53^flox/flox^* model was previously described [1].

Mice carrying conditional *Notch1* and *Notch2* alleles were obtained from The Jackson Laboratory (www.jax.org (accessed on 25 August 2023)) and crossed into the *Pten^flox/flox^ p53^flox/flox^* background to create an initial set of *Pten^flox/wt^ p53^flox/wt^ Notch(1 or 2)^flox/wt^* mice. In order to minimise confounding effects of the mixing of different background strains, both control cohorts with wild-type *Notch* alleles and cohorts heterozygous and homozygous for conditional *Notch* and *p53* alleles were bred from this first set of crosses heterozygous for all alleles. The full details of the animals reported here are provided in Appendix A.

### 2.2. Tumour Phenotyping

Mice were euthanised when tumours reached previously defined humane size limits. Tumours were dissected and fixed in an excess volume of 4% neutral-buffered formalin for 24 h at 4 °C, followed by paraffin embedding. When a tumour was of sufficient size and not substantially necrotic, a piece was also snap-frozen on dry ice at time of dissection and then stored at −80 °C for later RNA and protein extraction.

Histopathological analysis was carried out blinded to genotype by MJS (who has over ten years’ experience using the four-histotype classification system for mouse mammary tumours) using our previously established criteria [1,2,3] based primarily on morphology of haematoxylin–eosin (H&E)-stained sections and immunohistochemical staining for ΔNp63. For the latter, fresh sections were cut from FFPE tissue, dewaxed and re-hydrated. Sections underwent antigen retrieval in citrate buffer, pH 6.0 (Sigma, Gillingham, Dorset, UK), in a pressure cooker for 15 min before incubation with a 3% hydrogen peroxide solution for 20 min and then blocking in 10% goat serum/0.1% Tween-20/TBS for 1 h. Incubation with anti-ΔNp63 antibodies (ab735, Abcam, Cambridge, UK; 1:100) was performed overnight at 4 °C. Detection was carried out using an ImmPRESS kit (Vector Labs, Peterborough, UK). Sections were counterstained with haematoxylin and mounted. Images were acquired using an Olympus BX43 microscope.

### 2.3. Gene Expression Analysis Using Quantitative Real-Time rtPCR

For qRT-PCR analysis of gene expression, frozen tumour material was prepared using a Maxwell SimplyRNA LEV Tissue Kit for automated extraction of total RNA (Promega, Southhampton, Hampshire, UK). Briefly, a micro-pestle was used to grind frozen tumour material, on dry ice, prior to adding homogenisation buffer containing 1-Thioglycerol and an equal volume of lysis solution, and the relevant program used for automated RNA extraction with DNase I treatment. Samples were stored at −80 °C until used for cDNA synthesis, where 1 µg of RNA per sample was converted to cDNA using a Quantitect cDNA Synthesis Kit (Qiagen, Manchester, UK). qPCR reactions were performed using the cDNA as described previously [15]. Details of Taqman probes (Thermofisher, Paisley, UK) can be found in Appendix A. All results were calculated using the Δ–Δ_Ct_ method normalised to β-actin and expressed as mean fold gene expression difference over comparator samples with 95% confidence intervals.

Normal luminal estrogen receptor (ER) negative mammary epithelial cells isolated from 10-week-old virgin female C57Bl6 mice according to our previous protocols [2,15,16] were used as a normal comparator population. Three independent cell isolates were obtained. In brief, single cells were liberated from freshly isolated 4th mammary fat pads by a combination of mechanical and enzymatic digestion and then immediately stained with antibodies against CD45 (clone 30-F11, BD Biosciences, Oxford, UK, 1 µg/mL), CD24 (clone M1/69, BD Biosciences, 0.5 µg/mL) and Sca-1 (clone D7, BD Biosciences, 0.2 µg/mL), as well as DAPI. CD45^−^, CD24^+/High^ and Sca-1^−^ (luminal ER negative mammary epithelial cells, the cell population in which the Blg promoter drives Cre expression) [2] were isolated using flow cytometry, resuspended in RLT buffer (Qiagen, Crawley, West Sussex, UK) and stored at −80 °C until required for RNA extraction. Total RNA was extracted using an RNeasy MinElute Kit (Qiagen), according to the manufacturer’s instructions. cDNA synthesis and gene expression analysis were carried out as above.

Unsupervised hierarchical clustering of relative gene expression levels was carried out on Log2-transformed data using the Morpheus online tool (https://software.broadinstitute.org/morpheus/ (accessed on 25 August 2023)).

### 2.4. Western Blotting

To prepare samples for Western blotting, snap-frozen tissue was homogenised in RIPA buffer containing protease inhibitor cocktail (Roche, Burgess Hill, West Sussex, UK). The solution was then passed through a 23G needle and centrifuged for 10 min at 4 °C. The supernatant was centrifuged again, collected and stored at −80 °C. Protein extracts were separated using SDS-PAGE, transferred to PVDF membranes (IPVH00010, Merck Millipore, Hertfordshire, UK) and immunoblotted with antibodies against total AKT (#4685, Cell Signalling Technology, London, UK), phospho-S473-AKT (#9271, Cell Signalling Technology) or GAPDH (CB1001, Merck Millipore, Watford, UK) as the loading control. One common sample was run on every gel to provide a normalisation standard, enabling cross-comparison between experiments. Resulting immunocomplexes were detected by HRP-conjugated anti-mouse IgG or anti-rabbit IgG secondary antibodies as appropriate and enhanced chemiluminescent (ECL) reagents (WBLUF0100, Merck Millipore). Blots were exposed to film for a range of times to optimise the appearance of the bands.

Bands were quantified using ImageJ. The background value for each lane was subtracted from each ban; then, the phospho- and total-AKT values were normalised to the GAPDH value. The GAPDH-normalised pAKT was then normalised to the GAPDH-normalised total AKT to give a corrected value for AKT phosphorylation, allowing for both protein loading (normalisation to GAPDH) and different levels of AKT (normalisation to total AKT). These values were then normalised to the standard control on each blot to enable different gels to be compared.

### 2.5. Statistics

Significance of changes in distribution of tumour types was determined by a Chi^2^ test of distribution of categorical variables. For survival curves, the logrank test was used. ANOVA tests were used in all other cases. Significance of qrtPCR results was determined from 95% confidence intervals according to [17]. All statistical analysis was carried out using GraphPad Prism version 9.

## 3. Results

### 3.1. Notch1 and Notch2 Are Differentially Expressed in Mammary Tumours

Notch signalling is a key signalling pathway for normal development in epithelial tissues, including the mammary gland [7]. To determine whether differential expression of the mammalian Notch receptors NOTCH1 and NOTCH2, which are particularly associated with different breast cancer subtypes with different prognosis [11,12,13,14], is found in mouse mammary tumours of different histotypes and may, therefore, contribute to the generation of tumour heterogeneity, we analysed receptor expression using qrtPCR in tumours from two different genetic backgrounds (previously reported elsewhere) [1]. *BlgCre Pten* and *BlgCre Pten p53* tumours had similar levels of *Notch1* expression relative to normal mouse mammary luminal oestrogen receptor (ER)-negative progenitors (the cell of origin of tumours in animals carrying the *BlgCre* transgene) [2] (Figure 1A,B). In contrast, *BlgCre Pten* and *BlgCre Pten p53* tumours had significantly higher levels of *Notch2* expression relative to their cell of origin (Figure 1C,D; Appendix A).

### 3.2. Notch1/2 Deletion Does Not Alter Tumour Onset

To determine whether *Notch1/2* signalling is a regulator of tumour histotype, *BlgCre Pten p53* mice were crossed with mice carrying conditional (‘floxed’) *Notch1* or *Notch2* alleles. Cohorts were generated such that all mice carried homozygous floxed *Pten* alleles, as well as the *BlgCre* transgene. However, different cohorts had both heterozygous and homozygous floxed *p53* and *Notch* alleles in different combinations. Noteworthily, it proved difficult to generate homozygous conditional *p53* mice from the *BlgCre Pten p53 Notch1* line. The reason for this is unknown.

The presence of heterozygous or homozygous floxed *Notch* alleles made no difference to survival on the *Pten p53* background (Figure 2). However, the presence of homozygous floxed *p53* alleles in the *BlgCre Pten p53 Notch2* background resulted in a significantly (*p* < 0.001) shorter survival, irrespective of the *Notch2* allele status (Figure 2B,D). We were unable to breed sufficient homozygous *p53* animals in the *BlgCre Pten p53 Notch1* cohort to be able to robustly assess this finding in those cohorts. However, the homozygous *p53* animals we were able to generate survived for very similar times to their heterozygous *p53* counterparts (Figure 2A).

Therefore, the deletion of *Notch1/2* does not alter tumour onset in the *BlgCre Pten p53* background, but *p53* allele status can affect tumour onset, as has been previously demonstrated [18].

### 3.3. Notch Deletion Increases the Proportion of AME and ASQC Tumours

Tumour histotypes from the different cohorts were next analysed, using the same four categories (MSCC, ASQC, AME and AC(NST)) as previously [1,2,3]. These tumour types can be readily diagnosed from H&E and ΔNp63 staining; typical examples of the four types and key differential diagnostic features (primarily, the presence of metaplastic features and the number/pattern of p63 stained cells) are shown in Figure 3A–D. Cohorts were analysed by *Notch* allele status (wild-type, heterozygous or homozygous); the results from *p53* heterozygous and homozygous mice of the same *Notch* allele status were pooled. Note that if an individual animal had >1 tumour that could be analysed, the phenotypes of the multiple tumours were not necessarily the same (Appendix A).

In all cohorts, the presence of one or two conditional *Notch* alleles resulted in a dose-dependent increase in the proportion of ASQCs (although the difference between the *BlgCre Pten p53 Notch1^wt^* and the *BlgCre Pten p53 Notch1^het^* cohorts did not reach statistical significance). In the *Notch1* cohort, this appeared to be at the expense of MSCC and AC (NST) phenotypes, whereas in the *Notch2* cohort, the AME phenotype tumours were lost, while the MSCC tumours were retained (Figure 3E,F).

### 3.4. Expression of Lineage-Associated Genes Is Associated with Histotype Rather than Notch1/2 Status

Next, we assessed whether the effects of deletion of *Notch1* or *Notch2* had different effects on the patterns of gene expression, in particular, on genes associated with mammary epithelial lineages and NOTCH signalling. qrtPCR analysis of the tumours was carried out using a panel of genes associated with the three main mammary epithelial cell lineages (basal: *Fzd7*, *Id4*, *Jag1*, *Jag2*, *Krt14*, *Krt15*, *Notch4*, *Runx2* and *Tp63*; luminal ER negative: *Foxc1* and *Sox6*; luminal ER positive: *Esr1*, *Foxa1* and *Msx2*; and both luminal populations: *Cd24a*, *Notch1*, *Notch2*, *Notch3* and *Krt18*) [15] and with NOTCH signalling (*Dtx1*, *Fabp7*, *Fbxw7*, *Jag1*, *Jag2*, *Hes1*, *Hes2*, *Heyl*, *Notch1-4* and *Nrarp*).

Unsupervised hierarchical clustering of the tumours on the basis of the relative levels of expression of this gene set divided the samples into five tumour clusters and two gene clusters (Figure 4; Appendix A). Gene cluster A (‘differentiation cluster’) was composed of nine lineage-associated genes (82%) and two NOTCH-associated genes (18%), while gene cluster B (‘NOTCH cluster’) was composed of four lineage-associated genes (26%), six NOTCH-associated genes (40%) and five NOTCH genes (34%), which were also associated with particular cell lineages. Tumour cluster I had low levels of expression of genes from both gene clusters A and B and was composed of nine (75%) MSCCs and three (25%) ASQCs. Tumour cluster II was composed of a mix of tumour histotypes and had intermediate levels of expression of genes in cluster A but high levels of cluster B gene expression. Tumour cluster III was composed of mainly (six out of seven) MSCCs and had intermediate levels of cluster A expression and low levels of cluster B expression. Tumour clusters IV and V were composed mainly of ASQCs, and both had high levels of gene cluster A expression. Tumour cluster IV had intermediate gene cluster B expression, while tumour cluster V had high levels of expression of the NOTCH gene cluster. *Notch* wild-type tumours were scattered throughout the data set, and there was no obvious clustering of the *Notch2* knockout tumours. The *Notch1* knockout tumours were particularly enriched among tumours from tumour clusters IV and V, consistent with the depletion of MSCC phenotype tumours from this cohort (Figure 3E) and the enrichment of tumour clusters I and III with tumours of this histotype. Overall, it appeared that, at least using this set of genes, biological similarities and differences between tumours were more strongly influenced by the expression of markers associated with lineage and cell differentiation (gene cluster A) rather than NOTCH signalling (gene cluster B). The expression of markers associated with lineage and cell differentiation was associated with histotype rather than the *Notch1* or *2* status of a tumour.

### 3.5. AKT Signalling Is Upregulated in AME and ASQC Tumours

Both the AME and ASQC phenotypes are characterised by stereotypical patterns of expression of ΔNp63 (Figure 3), a transcriptional regulator important for the function of the basal cell layer of stratified epithelia. ΔNp63 is typically expressed only in the basal (myoepithelial) layer of the mammary epithelium in the resting (non-pregnant) gland, although it can be observed in luminal cells during pregnancy when the mammary epithelium is proliferating to increase tissue mass ready for lactation [19]. However, we observed in some apparently dysplastic/early pre-neoplastic ducts adjacent to the tumours in our mouse models that ΔNp63 was expressed in luminal as well as basal cells (i.e., not in the typical pattern associated with the non-pregnant gland). The more typical, basal-only pattern could also be observed (Figure 5A,B; Appendix A). Unfortunately, only a limited number of samples contained tumour-adjacent ducts, which meant any differences between genotypes in the number of luminal cells stained for ΔNp63 could not be reliably quantified.

We recently demonstrated that in a mouse model of mammary cancer driven by a conditionally activated *Erbb2/HER2* orthologue, the ASQC phenotype is associated with the activation of the epidermal differentiation cluster of genes (EDC), and we suggested that this was dependent on a ΔNp63-regulated super enhancer [3]. ΔNp63 is regulated by both NOTCH [20] and PI3K–AKT signalling [21]; PTEN is a well-established regulator of PI3K activity. PI3K pathway activation has previously been associated with lineage switching (luminal to basal and basal to luminal) in mouse mammary epithelium [22]. We, therefore, hypothesised that a key determinant of differentiation to the ASQC phenotype would be high levels of PI3K-AKT activity. To test this, Western blot analysis of tissue from the range of tumour phenotypes was carried out to measure the active phosphorylated form of AKT, pS473. When tumours were grouped by NOTCH status (wild-type, *Notch1* knockout and *Notch2* knockout), there was no difference in levels of activated AKT (Figure 5C). However, when tumours were grouped by phenotype, AME tumours had significantly higher levels of pAKT (*p* < 0.05), and ASQC tumours had highly significantly increased (*p* < 0.001) levels of pAKT, consistent with the model (Figure 5D; Appendix A). Therefore, strong activation of the PI3K–AKT signalling pathway is associated with specific tumour phenotypes, but not *Notch* allele status, in this model system.

## 4. Discussion

Inter-tumour heterogeneity is associated with recurrent features that enable tumours to be classified into distinct recognisable categories that can have predictive/prognostic value. For example, a comedo growth pattern is a poor prognostic feature in ductal carcinoma in situ of the human breast [23] and also in canine mammary cancer [24]. Interestingly, while comedocarcinoma of the canine mammary gland has a high risk of distant metastasis, it has a low rate of local recurrence following therapy; in contrast, adenosquamous canine mammary tumours have a very high rate of local recurrence [24]. These recurrent features suggest stereotypical non-random processes, which may be termed ‘tumour developmental biology’ [5]. Here, we have developed a model of tumour developmental biology that, taken together with our previous findings [1,2,3], explains the origins of the tumour histotypes seen in a simple inter-tumour heterogeneity model system. We find that the deletion of either *Notch1* or *Notch2* in the *BlgCre Pten p53* background resulted in a dose-dependent increase in ASQC tumours, although *Notch2* deletion is also associated with the MSCC phenotype. Furthermore, AME/ASQC tumour histotypes were associated with an increase in activated AKT (as defined by the presence of pS473 AKT). Therefore, active PI3K/AKT signalling is associated with the ASQC and AME phenotypes, while the NOTCH pathway is an ASQC suppressor. As both of these tumour types are associated with high expression of the basal transcription factor ΔNp63, this is consistent with previous findings that NOTCH activation cell-autonomously dictates luminal cell fate specification in mammary epithelial cells [8]. However, there are clearly NOTCH-family-receptor-specific effects as well, as deletion of *Notch1* or *Notch2* had slightly different effects on tumour phenotype (Figure 3E,F) and tumour gene expression (Figure 4), in particular, the number of MSCC tumours, suggesting a role of NOTCH2 in the regulation of EMT; links between NOTCH and EMT have been previously suggested [25].

We define the AME histotype as a tumour that contains distinct neoplastic glandular ΔNp63-negative and pseudo-basal ΔNp63-positive populations arranged in a stereotypical architecture, while the ASQC histotype is typified by the presence of nests of ΔNp63 positive cells undergoing squamous metaplasia, frequently surrounding keratin pearls (Figure 3). However, ASQC tumours frequently contain regions with an AME-like pattern of ΔNp63 staining, suggesting AME and ASQC tumours are on a spectrum of inter-tumour heterogeneity. The presence of large numbers of ΔNp63-positive cells is a key feature these histotypes have in common, and we propose that ΔNp63 is the key driver of tumour developmental biology in this model system. We have recently shown that the EDC gene cluster of keratinocyte-associated differentiation genes is upregulated in ASQC tumours in vivo and suggested a role for p63 in the activation of the EDC in these tumours [3,26]. It is notable that in an in vitro culture system, strong activation of both WNT and ERBB2 pathways (the latter also activating PI3K–AKT signalling) also resulted in squamous metaplasia in mammary epithelial organoids [27], while the expression of a mutant active PI3K in the luminal cells of the mammary epithelium resulted in the movement of cells into the basal layer [22].

ΔNp63 is required for the formation of stratified epithelia [28,29,30] and expressed in basal epithelial cells, including the basal stem cell layer of the epidermis and the basal/myoepithelial cells of the resting (non-pregnant) mammary gland [19,31]. P63 expression is regulated by a network of well-known developmental pathways (in particular, NOTCH, WNT, Hedgehog, FGFR2 and EGFR signalling), often with complex negative and positive feedback loops, characteristics of systems that establish and maintain tissue boundaries (reviewed in [32]). EGFR signalling in keratinocytes induced ΔNp63 expression through PI3K signalling [21] (which is negatively regulated by PTEN), while NOTCH represses ΔNp63 expression [20]. We suggest that our model system enabled ΔNp63 to be induced in luminal cells in a dose-dependent manner following the knockout of PTEN and NOTCH, leading to luminal-to-basal metaplasia and the AME tumour phenotype. Concurrent (or subsequent) activation of the epidermal differentiation cluster of genes would then drive ASQC formation (Figure 5E). However, this hypothesis remains to be formally demonstrated, as it would require the introduction of conditional knockout *Tp63* alleles into the mouse lines we describe here to determine whether *Tp63* knockout can rescue the shift to the AME/ASQC phenotypes that result from *Notch1/2* knockout. It is notable that in the *Notch2* knockout cohorts, this driving of tumour development towards the formation of a highly specialised epithelial cell type, squamous epithelium, happened alongside other tumours that were being driven to lose all epithelial features and undergo EMT to become spindle cell tumours. The loss of *Notch2* may have meant the loss of a feedback loop that acts as a break upon both differentiation and de-differentiation, whereas *Notch1* only acts to regulate differentiation.

Our study has limitations. The complexity of the animal breeding programme meant that cohorts were established over an extended period of time, and we cannot exclude that genetic drift may have occurred within each line. However, the cohorts were collected over similar timeframes so the results within each genetic background are compatible. Furthermore, there is no obvious bias of one particular histotype towards animals added to a cohort study early or late, which argues against a significant genetic drift effect. We also acknowledge that as our study was on in vivo tissue samples taken at point of euthanasia, we were unable to determine how tumour histotype may evolve as a tumour grows. We also did not assess visceral metastasis.

We were not able to directly confirm *Notch1/2* deletion in end-stage tumours. We were not able to access isoform-specific NOTCH antibodies for use in immunohistochemistry, and we have yet to identify markers that enable us to purify mouse mammary tumour cells for analysis (e.g., by flow cytometry) that are specific for the tumour and would not also isolate normal cells. This also meant that for the gene expression analysis, we analysed whole tumour pieces that could potentially have included non-recombined non-transformed cells (e.g., tumour-associated fibroblasts or normal mammary ducts) trapped within the tumour (although we did ensure that no visibly normal tissue on the outside of a piece of tumour was included in this processing). Importantly, however, the fact that the shift in tumour phenotypes is ‘dose-dependent’, i.e., the effect becomes stronger depending on whether the genotypes are *Notch^wt/wt^*, *Notch^flox/wt^* or *Notch^flox/flox^*, is strong, although indirect, evidence that there is efficient recombination of the conditional *Notch* alleles in the tumour cells.

## 5. Conclusions

We find that in mouse models of mammary epithelial tumour histotype heterogeneity, the NOTCH pathway is a suppressor of the ASQC histotype, while PI3K/AKT signalling enhances both AME and ASQC. We propose these pathways act through ΔNp63. With the addition of the epithelial–mesenchymal transition as a mechanism driving the formation of metaplastic spindle cell tumours, a process which may also be regulated by NOTCH2 [33], a model of the differentiation pathways that drive mammary tumour heterogeneity can begin to be built (Figure 5E). Given that the mammary gland, salivary gland and other adnexal glands have similar developmental origins, and that ΔNp63 is a diagnostic feature in adenomyoepitheliomas and squamous tumours in multiple tissues, this model likely has general applicability. It is also the first step in understanding growth patterns such as comedo necrosis, papillary growth, etc., which will require, we would argue, the application of systems biology approaches to integrate differentiation pathway models, such as the one we propose here, with proliferation, apoptosis and interactions between tumour and normal tissue, to establish comprehensive in silico three-dimensional models of tumour developmental biology.

## Figures and Tables

**Figure 1 cancers-15-04324-f001:**
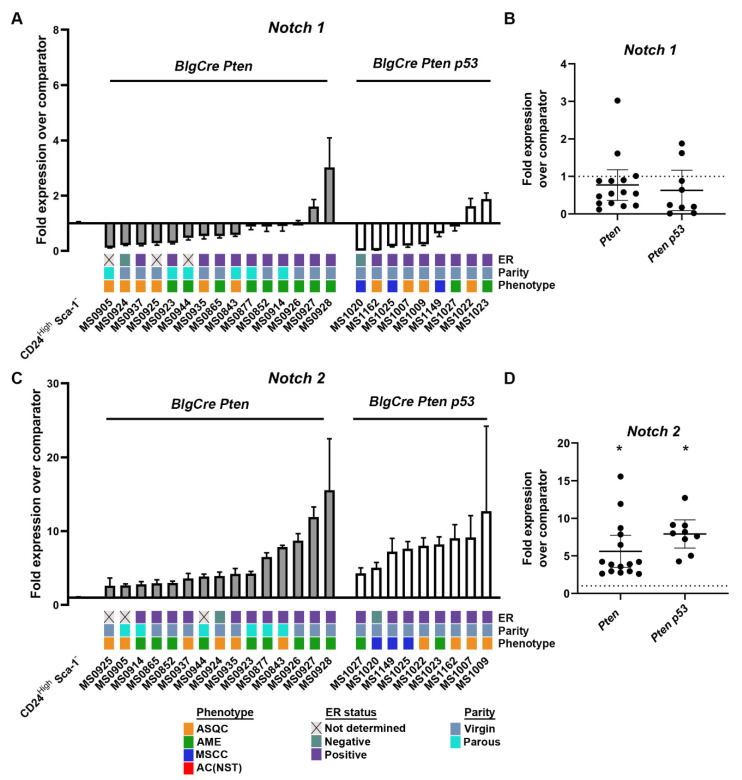
*Notch1* and *Notch2* are differentially expressed in *Brca1* and *Pten* conditional deletion mouse mammary tumour models. Quantitative real-time rtPCR (qrtPCR) expression analysis of *Notch1* (**A**,**B**) and *Notch2* (**C**,**D**) in mammary tumours (means ± 95% confidence intervals from three technical replicate analyses of each tumour) from *BlgCre Pten* and *BlgCre Pten p53* genetically engineered mouse tumour models [1] relative to expression in a normal comparator population, purified luminal ER negative mammary epithelial stem/progenitor cells (CD24^High^ Sca-1^−^ cells), which are the cell of origin for *BlgCre*-driven tumours [2]. A and C show tumour-by-tumour results. The oestrogen receptor (ER) status, parity of the animal and the histological phenotype of the tumours, as previously published [1], are indicated below each tumour. Numbers starting with ‘MS’ are tumour-specific identifiers. B and D show summarised relative expression values for each gene. Means ± 95% CI are indicated. * *p* < 0.05, as determined from 95% confidence intervals according to [16].

**Figure 2 cancers-15-04324-f002:**
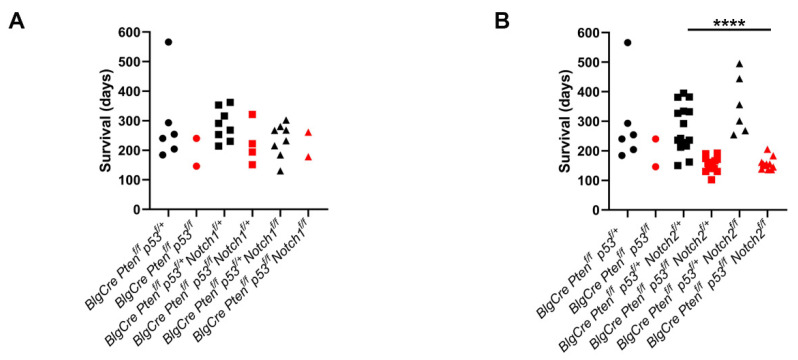
Deletion of *Notch1* or *Notch2* does not accelerate tumour formation. (**A**,**B**) Age at which mice were euthanised due to specified endpoints being reached in *Pten p53 Notch1* (**A**) and *Pten p53 Notch2* (**B**). Cohorts with heterozygous or homozygous *p53* and *Notch* alleles are shown separately. *Notch* wild type cohorts shown as circles, *Notch* heterozygous cohorts shown as squares, *Notch* homozygous cohorts shown as triangles. Heterozygous *p53* cohorts are plotted in black, homozygous *p53* cohorts are in red. It proved difficult to generate large numbers of animals with homozygous floxed *p53* alleles in the *Pten p53 Notch1* cohorts; however, the animals that were generated had comparable survival to p53 heterozygous mice (**A**). Homozygous floxed *p53* animals on the *Notch2* background, however, had significantly reduced survival compared to heterozygous *p53* animals on the same background (**B**) (**** *p* < 0.0001, ANOVA). (**C**) Survival curve for *BlgCre Pten^f/f^ p53^f/+^*, *BlgCre Pten^f/f^ p53^f/+^ Notch1^f/+^* and *BlgCre Pten^f/f^ p53^f/+^ Notch1^f/f^* cohorts. No significant difference in survival between cohorts. Homozygous *p53* cohorts not plotted due to low numbers. (**D**) Survival curve for *BlgCre Pten^f/f^ p53^f/+^* (same data as in (**C**)), *BlgCre Pten^f/f^ p53^f/+^ Notch2^f/+^*, *BlgCre Pten^f/f^ p53^f/+^ Notch2^f/f^*, *BlgCre Pten^f/f^ p53^f/f^ Notch2^f/+^* and *BlgCre Pten^f/f^ p53^f/f^ Notch2^f/f^* cohorts. On the *Pten Notch2* background, the shift from heterozygous to homozygous *p53* alleles caused a significant reduction (**** *p* < 0.0001, logrank test) in survival as a result of mammary tumour development. However, the presence of either heterozygous or homozygous *Notch2* alleles had no effect on mammary-tumour-specific survival.

**Figure 3 cancers-15-04324-f003:**
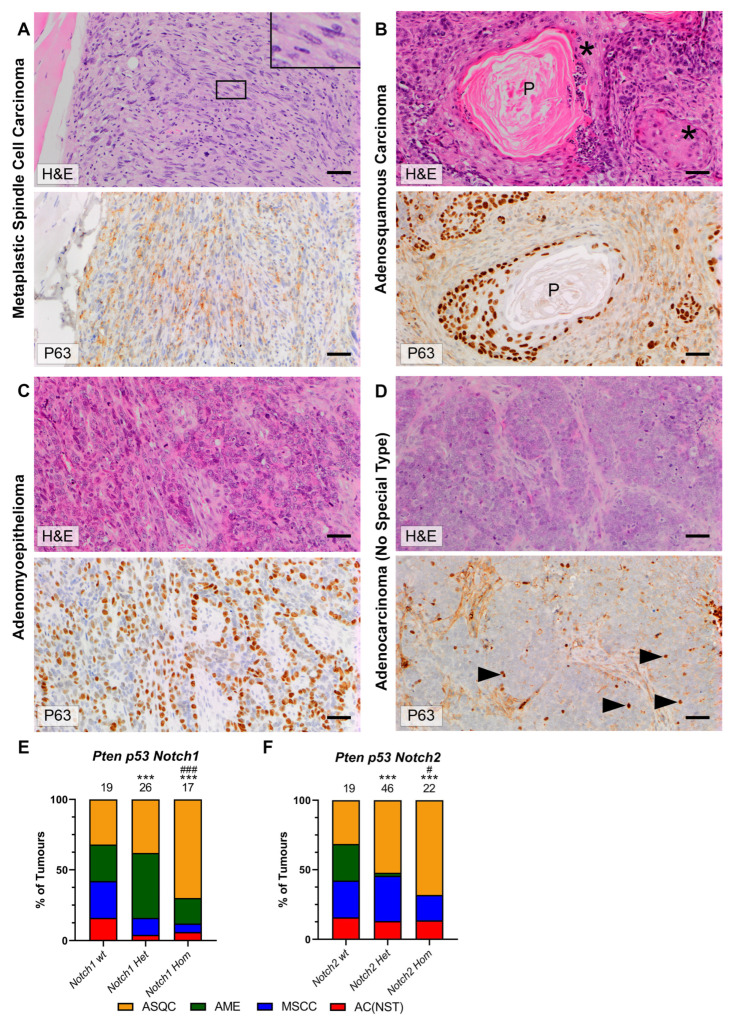
Notch deletion alters tumour phenotype. (**A**–**D**) H&E and ΔNp63 immunohistochemical staining of tumours representative of the four typical histotypes showing key differential diagnostic features. Bars = 50 μm. (**A**) Mesenchymal spindle cell carcinoma (MSCC) consisting of a majority of tightly packed fusiform cells with elongated nuclei (inset). In addition, these tumours show little or no staining for cytokeratins (not shown); background cytoplasmic staining is present in the lower panel. (**B**) Adenosquamous carcinoma (ASQC) with nests of cells undergoing squamous metaplasia (asterisk) and keratin pearls (P). Intense ΔNp63 nuclear staining in areas of squamous change and around keratin pearls. (**C**) Adenomyoepithelioma (AME) consisting of distinct neoplastic glandular p63 negative and pseudo-basal nuclear ΔNp63-positive populations. (**D**) Adenocarcinoma of no special type (AC(NST)) composed of sheets and nests of cuboidal cells embedded within the tissue stroma. Moderate to strong nuclear pleomorphism with occasional ΔNp63 nuclear positivity (arrowheads). (**E**,**F**) Percentages of the four different tumour histotypes in *Pten p53 Notch1* (**E**) and *Pten p53 Notch1* mice (**F**). Total number of tumours analysed in each cohort is shown above the bars. Cohort data are split according to *Notch* conditional allele status (wild-type, heterozygous or homozygous floxed). Results from *p53* heterozygous and homozygous animals are considered together. *** *p* < 0.001 vs. *Notch* wt cohort; # *p* < 0.05, ### *p* < 0.001 vs. *Notch* het cohort; Chi^2^ test of proportion of categorical variables.

**Figure 4 cancers-15-04324-f004:**
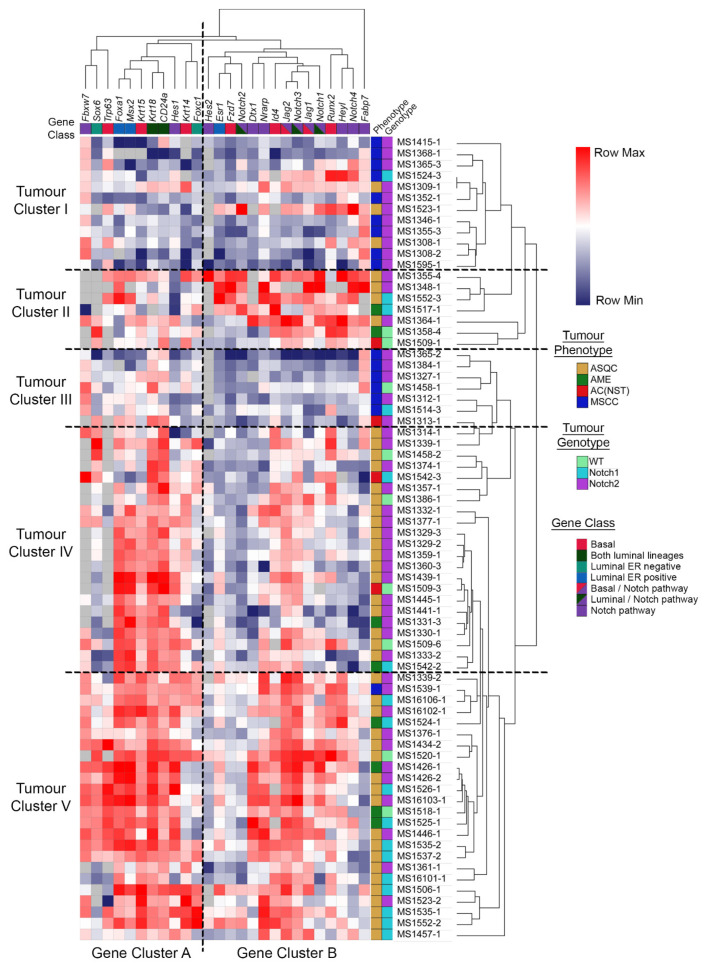
Expression of lineage-associated genes is associated with histotype. Unsupervised hierarchical clustering of expression of 26 genes associated with basal and luminal differentiation and NOTCH signalling in mouse mammary tumours. Gene expression was determined by fold change relative to a comparator (tumour 1309-1) and Log2-transformed prior to unsupervised hierarchical clustering to identify both groups of tumours and groups of genes with similar expression patterns. Left-hand-side keys indicate tumour genotype, tumour phenotype and gene class. The latter is based on whether a gene is either a known NOTCH signalling component or was identified as being most strongly expressed in basal mammary epithelial cells, all luminal mammary epithelial cells or the luminal-ER-positive or ER-negative subpopulations in our previous studies [15].

**Figure 5 cancers-15-04324-f005:**
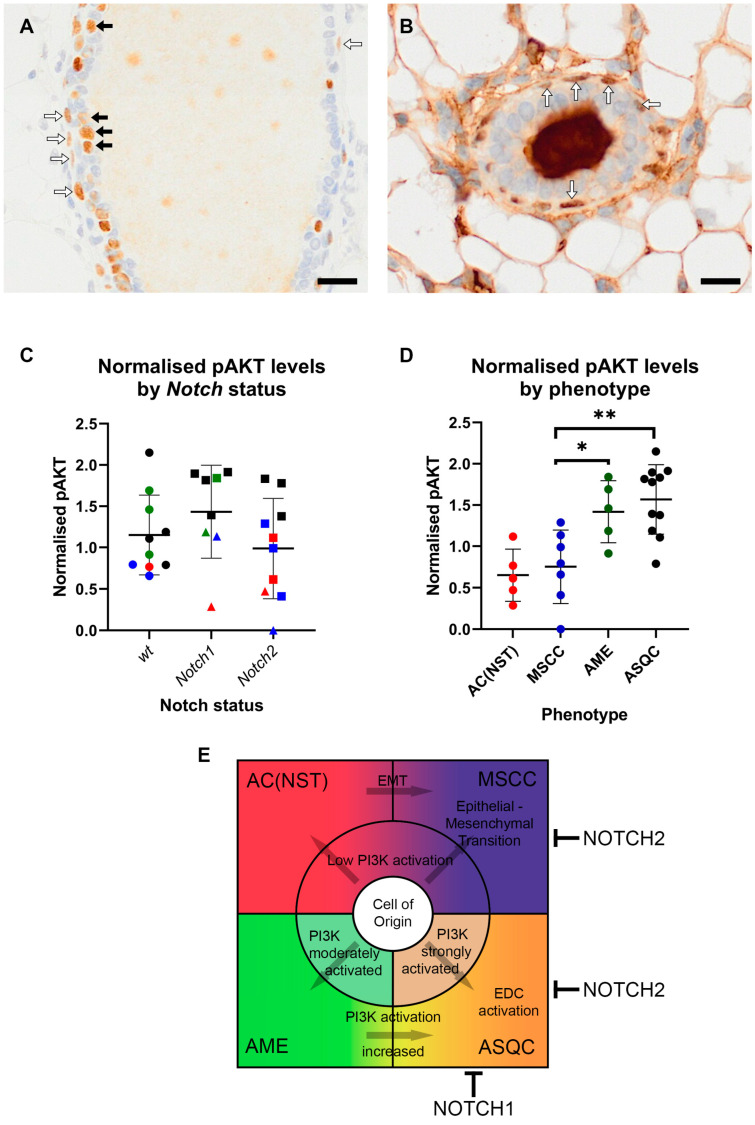
ASQC and AME tumours have significantly higher levels of phosphorylated AKT. (**A**,**B**) Examples of ΔNp63-positive cells in mammary ducts adjacent to tumours ((**A**) MS1628-1, *Notch 1^flox/+^*; (**B**) MS1520-1, *Notch* wild-type). White arrows (**A**,**B**) indicate examples of positive cells with elongated nuclei lying flat against the duct at the outer periphery, consistent with myoepithelial cells. Black arrows (**A**) indicate examples of cuboidal to columnar cells in a suprabasal or luminal position, consistent with a luminal epithelial cell identity but showing ‘atypical’ p63 staining. Bars = 25 μm. (**C**,**D**) Levels of phospho-S473 AKT normalised to total AKT, GAPDH and common control sample in mouse mammary tumours. (**C**) No significant difference in normalised pAKT in tumours grouped by *Notch* status. Each symbol is an individual tumour. Symbol colours indicate tumour phenotype (red, AC(NST); blue, MSCC; green, AME; black, ASQC); symbol shapes indicate *Notch* allele status (circle, wt; triangle, heterozygous; square, homozygous). (**D**) Grouping normalised pAKT values by tumour phenotype demonstrates significantly higher levels of pAKT in AME (* *p* < 0.05, ANOVA) and, in particular, ASQC tumours (** *p* < 0.01, ANOVA) compared to AC(NST) and MSCC tumours. Raw blots, the bands used for quantitation, the ImageJ data and the calculations are provided in Appendix A. (**E**) Model of tumour histotype development in the mouse mammary epithelium. EDC—epidermal differentiation cluster. EMT—epithelial–mesenchymal transition.

## Data Availability

Relevant raw data are provided in the Appendix A. No large data sets were generated during this study.

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
