# Peer review of "NOTCH and AKT Signalling Interact to Drive Mammary Tumour Heterogeneity"

_cancers, 2023, doi:10.3390/cancers15174324_

Round 1

Reviewer 1 Report

This is an interesting manuscript that tests the effects of deleting NOTCH1 and NOTCH2 on mammary tumor formation using genetically engineered tumor models. The authors provide data demonstrating expression of NOTCH1 and NOTCH2 in mammary tumors from BlgCre Pten p53 and Blg Pten mice in comparison with normal mouse mammary luminal ER negative progenitors. To determine the functions of NOTCH1 and NOTCH2 in tumor formation and growth, mice were generated using Notch1 and Notch2 floxed mice and effects on tumor growth and histology was assessed. While deletion of Notch1 or 2 does not affect survival, there are dose dependent changes in the types of tumors that are formed in these mice. Specifically ,the results demonstrate that the NOTCH pathway suppresses a metaplastic adenosquamous carcinoma histotype. Notably, this histotype is associated with increased levels of pAKT, although NOTCH status was not found to be directly associated with pAKT levels. The findings in these studies convincingly suggest a role for the NOTCH pathway in regulating mammary tumor histotype.

Comments:

How do the four histotypes correlate with human breast cancer? While this has presumably been covered in previous publications, some inclusion of this information within this manuscript would help provide important context for readers.

Figure 1: How was the “comparator” RNA obtained?

Figure 2: Did the authors confirm Notch1 and Notch2 deletion in end stage tumors?

Figure 2: Do these tumors metastasize and if so, was metastasis assessed in these mice?

Figure 3: It would be helpful to readers to include wording in the figure that identifies each tumor histotype in panels A-D.

Figure 5 A,B: This figure should include a duct from a normal mammary gland as a control to indicate what is meant by “ectopic expression” in the genetically engineered models. Also, the text states that this was seen in all genotypes although images from only one genotype appears to be shown, other images could be included as supplemental data.

At the end of the Introduction, the authors suggest a model by which PI3K/AKT and NOTCH combine to regulate differentiation pathways in mammary tumors via DNp63 and they also hypothesize that cellular signaling pathways converging on DNp63 drive tumors towards the ASQC phenotype (lines 318-319). However, data related to DNp63 are limited to correlation with tumor histotype. Thus, this hypothesis is not explicitly tested in this study. The authors should carefully review the wording of their conclusions, especially related to DNp63.

Reviewer 2 Report

In this study using genetically modified mouse models, the deletion of NOTCH signaling pathway genes resulted in an increase in metaplastic adenosquamous carcinomas (ASQCs) and adenomyoepitheliomas (AMEs), accompanied by elevated PI3K/AKT signaling, suggesting the involvement of these pathways in determining tumor histotype is interesting. However, the development of new approaches enhances the interest in the field. The manuscript flows well. No changes required here.  

Author Response

This reviewer requires no changes. We are grateful for the supportive comments.

Round 2

Reviewer 1 Report

Tha authors have addressed all comments, no further concerns.